

# Integrated microRNA, gene expression and transcription factors signature in papillary thyroid cancer with lymph node metastasis

Nurul-Syakima Ab Mutalib[1,*], Sri Noraima Othman[1,*], Azliana Mohamad Yusof[1], Shahrun Niza Abdullah Suhaimi[2], Rohaizak Muhammad[2] and Rahman Jamal[1]

[1] UKM Medical Molecular Biology Institute, Universiti Kebangsaan Malaysia, Cheras, Kuala Lumpur, Malaysia
[2] Department of Surgery, Faculty of Medicine, Universiti Kebangsaan Malaysia, Cheras, Kuala Lumpur, Malaysia
[*] These authors contributed equally to this work.

Corresponding author
Nurul-Syakima Ab Mutalib,
syakima@ppukm.ukm.edu.my

## ABSTRACT

**Background**. Papillary thyroid carcinoma (PTC) is the commonest thyroid malignancy originating from the follicle cells in the thyroid. Despite a good overall prognosis, certain high-risk cases as in those with lymph node metastasis (LNM) have progressive disease and poorer prognosis. MicroRNAs are a class of non-protein-coding, 19–24 nucleotides single-stranded RNAs which regulate gene expression and these molecules have been shown to play a role in LNM. The integrated analysis of miRNAs and gene expression profiles together with transcription factors (TFs) has been shown to improve the identification of functional miRNA-target gene-TF relationships, providing a more complete view of molecular events underlying metastasis process.

**Objectives**. We reanalyzed The Cancer Genome Atlas (TCGA) datasets on PTC to identify differentially expressed miRNAs/genes in PTC patients with LNM-positive (LNM-P) versus lymph node negative (LNN) PTC patients and to investigate the miRNA-gene-TF regulatory circuit that regulate LNM in PTC.

**Results**. PTC patients with LNM (PTC LNM-P) have a significantly shorter disease-free survival rate compared to PTC patients without LNM (PTC LNN) (Log-rank Mantel Cox test, $p = 0.0049$). We identified 181 significantly differentially expressed miRNAs in PTC LNM-P versus PTC LNN; 110 were upregulated and 71 were downregulated. The five topmost deregulated miRNAs were hsa-miR-146b, hsa-miR-375, hsa-miR-31, hsa-miR-7-2 and hsa-miR-204. In addition, 395 miRNAs were differentially expressed between PTC LNM-P and normal thyroid while 400 miRNAs were differentially expressed between PTC LNN and normal thyroid. We found four significant enrichment pathways potentially involved in metastasis to the lymph nodes, namely oxidative phosphorylation (OxPhos), cell adhesion molecules (CAMs), leukocyte transendothelial migration and cytokine–cytokine receptor interaction. OxPhos was the most significantly perturbed pathway ($p = 4.70E-06$) involving downregulation of 90 OxPhos-related genes. Significant interaction of hsa-miR-301b with HLF, HIF and REL/NFkB transcription factors were identified exclusively in PTC LNM-P versus PTC LNN.

**Conclusion**. We found evidence of five miRNAs differentially expressed in PTC LNM-P. Alteration in OxPhos pathway could be the central event in metastasis to the lymph node in PTC. We postulate that hsa-miR-301b might be involved in regulating LNM in PTC via interactions with HLF, HIF and REL/NFkB. To the best of our knowledge, the roles of these TFs have been studied in PTC but the precise role of this miRNA with these TFs in LNM in PTC has not been investigated.

## INTRODUCTION

Papillary thyroid carcinoma (PTC) is the most common malignancy originating from the thyroid. Although the prognosis of PTC is generally good with a high 5-year survival rate, cases demonstrating certain clinicopathological parameters are progressive, have poorer prognosis and are considered as high-risk (*Ito et al.*, *2009*). Numerous classification systems for thyroid carcinoma have been established in order to classify high-risk cases such as AMES (*Cady & Rosai*, *1988*), AGES (*Hay et al.*, *1987*), MACIS (*Hay et al.*, *1993*) as well as TNM (*Sobin & Wittekind*, *2002*; *AJCC*, *2010*). The TNM classification is the most recent classification system and is based on size and extrathyroid extension (T), lymph node involvement (N), distant metastasis (M) and patient's age.

MicroRNAs (miRNAs), firstly identified in *Caenorhabditis elegans*, are a class of endogenous (non-protein-coding), 19–24 nucleotides single-stranded RNAs that derive from a stem-loop precursor to inhibit gene expression by binding primarily to the 3′-UTR of specific 'target' messenger RNA (mRNAs). MiRNAs that bind with perfect or nearly perfect complementarity to protein-coding mRNA sequences induce the RNA-mediated interference (RNAi) pathway, resulting in the disruption of mRNA stability and/or translation (*Bartel*, *2009*). Dysregulation of miRNAs expression in human cancers have been demonstrated by many studies (*Iorio & Croce*, *2012*). Through expression profiling studies, miRNAs were shown to be linked to tumor development, tumor progression, and response to treatment, signifying their potential use as biomarkers for diagnosis and prognosis (*Iorio & Croce*, *2012*). MiRNAs have also been shown function as biomarkers in predicting lymph node metastasis (LNM). There was a positive correlation between high hsa-miR-21 expression with tumor stage and LNM in patients with breast cancer (*Yan et al.*, *2008*), and the development of distant metastases in colorectal cancer patients (*Slaby et al.*, *2007*). Most recently, hsa-miR-1207-5p was suggested as a useful biomarker in the prediction of LNM in gastric cancer (*Huang et al.*, *2015*) and head and neck cancer (*De Carvalho et al.*, *2015*).

The current approach of miRNA target gene prediction via *in silico* analysis is built upon sequence similarity search and thermodynamic stability (*Alexiou et al.*, *2009*). Nevertheless, it is acknowledged that the results of *in silico* target prediction algorithms suffer from very low specificity (*Alexiou et al.*, *2009*). The combination of *in silico* target predictions with miRNA and gene expression profiles has been proven to improve the identification of

functional miRNA-target gene relationships (*Nunez-Iglesias et al.*, *2010*; *Ma et al.*, *2011*). As miRNAs act prevalently through degradation of the target genes, expression profiles of miRNA and target genes/transcripts are predicted to be inversely correlated (*Bisognin et al.*, *2012*). Another regulatory component, the transcription factors (TF), has also been shown to activate or repress miRNA expression level, further adding to the complexity of gene regulation. Efforts have been made to comprehend the mechanism of miRNAs in decreasing target genes expression; however the study of miRNA regulation by TFs (TF–miRNA regulation) is rather limited (*Wang et al.*, *2010a*).

The Cancer Genome Atlas (TCGA) Research Network recently published a molecular characterization of 507 PTCs and 59 matched normal adjacent tissues with respect to genomic, transcriptomic and proteomic signatures together with DNA methylation profiles, clinical and pathological features (*Cancer Genome Atlas Research Network*, *2014*). Data were collected through several studies across different institutions, thus creating a comprehensive dataset of PTC samples. Through unsupervised clustering methods, TCGA yielded six subtypes for miRNA expression and five for gene expression. However, miRNA and gene expression profiles between PTC with and without LNM were not comprehensively discussed. Here we reanalyzed these TCGA datasets on PTC with the aim of identifying differentially expressed miRNAs/genes in PTC patients with LNM-positive (LNM-P) as compared to lymph node negative (LNN) PTC patients and to investigate the miRNA-gene-TF regulatory circuit that governs LNM in PTC.

## MATERIALS AND METHODS

### TCGA papillary thyroid cancer dataset

We used the TCGA-generated microRNA sequencing (miRNAseq) and mRNAseq data for 495 tumors and 59 normal thyroid samples (*Cancer Genome Atlas Research Network*, *2014*). Metadata containing clinical information including *BRAF* V600E mutation status was obtained from cBioPortal (http://www.cbioportal.org/study.do?cancer_study_id=thca_tcga_pub#clinical) while miRNAseq and mRNAseq of 507 PTC patients were obtained from the TCGA Data Portal (https://tcga-data.nci.nih.gov/tcga/dataAccessMatrix.htm) (accessed from March 27, 2015 to May 25, 2015). Information were available for 507 PTC patients. The list of patients from the metadata was then filtered for PTC patients with N0, N1, N1a, and N1b, resulting in a total of 421 PTC patients out of the 507 patients (86 patients were excluded due to unavailability of node status). The clinical parameters are presented in Table 1.

Only samples with paired miRNAseq and mRNAseq data were selected, resulting in exclusion of additional three patients. In the end, we obtained a total of 418 patients' dataset which includes 213 patients with PTC LNN (N0) and 205 PTC LNM-P (53 patients with N1, 86 patients with N1a, 66 patients with N1b) (Table S1). Combined with 59 normal thyroid tissues, the total of datasets included in this study were 477. The miRNA and gene expression datasets consisting of 1,046 human miRNAs and 20,531 genes, respectively, were used for subsequent analysis.

**Table 1  Patient characteristics and integrated profiles in the TCGA PTC cohort.**

| Variables | PTC LNN | PTC LNM-P | | |
|---|---|---|---|---|
|  | N0 ($n = 213$) | N1 ($n = 53$) | N1a ($n = 86$) | N1b ($n = 66$) |
| Age range (years) | 15–85 | 19–83 | 18–83 | 19–89 |
|    Mean age | 49.4 | 41.9 | 43.5 | 48.4 |
| Gender ($n$) |  |  |  |  |
|    Male | 50 (23.5%) | 14 (26.4%) | 25 (29.1%) | 27 (40.9%) |
|    Female | 163 (76.5%) | 39 (73.6%) | 61 (70.9%) | 39 (59.1%) |
| Disease free status |  |  |  |  |
|    Recurred/progressed | 5 (2.3%) | 7 (13.2%) | 6 (7%) | 6 (9.1%) |
|    Disease free | 178 (83.6%) | 41 (77.4%) | 75 (87.2%) | 47 (71.2%) |
|    Unknown | 30 (14.1%) | 5 (9.4%) | 5 (5.8%) | 13 (19.7%) |
| Disease free (range in months) | 0.03–155 | 0–131 | 0–157 | 0.2 –46 |
|    Mean disease-free survival | 23.6 ($n = 183$) | 34.5 ($n = 48$) | 21.5 ($n = 81$) | 13.5 ($n = 53$) |
| Overall survival status |  |  |  |  |
|    Deceased | 35 (16.4%) | 12 (22.6%) | 11 (12.8%) | 19 (28.8%) |
|    Alive | 178 (83.6%) | 41 (77.6%) | 75 (87.2%) | 47 (71.2%) |
| Overall survival (range in months) | 0.03–155 | 0–131 | 0–157 | 0.2 –97.7 |
|    Mean overall survival | 24.3 ($n = 182$) | 35.2 ($n = 43$) | 21.4 ($n = 75$) | 15.2 ($n = 50$) |
| Extrathyroidal extension |  |  |  |  |
|    None | 160 (75.1%) | 31 (58.5%) | 49 (57%) | 37 (56.1%) |
|    Minimal (T3) | 42 (19.7%) | 14 (26.4%) | 33 (38.4%) | 23 (34.8%) |
|    Moderate/advanced (T4a) | 3 (1.4%) | 5 (9.4%) | 1 (1.2%) | 4 (6.1%) |
|    Very advanced (T4b) | 0 (0%) | 1 (1.9%) | 0 (0%) | 0 (0%) |
|    Unknown | 8 (3.8%) | 2 (3.8%) | 3 (3.5%) | 2 (3%) |
| BRAF status |  |  |  |  |
|    Mutated | 94 (44.1%) | 25 (47.2%) | 53 (61.6%) | 32 (48.9%) |
|    Wild type | 119 (55.9%) | 28 (52.8%) | 33 (38.4%) | 34 (51.5%) |

## Survival analyses

Kaplan–Meier survival analysis was carried out on disease-free and overall survival duration of TCGA PTC patients for whom follow-up details were available. Overall survival is defined as the duration from the date of diagnosis to death (due to all causes) while disease-free survival is defined as the duration from the date of the diagnosis to the date of recurrence, second cancer, or death due to all causes (whichever occurred first) (*Schvartz et al.*, *2012*). Curves were compared by univariate (log-rank) analysis. Statistical analyses were performed using GraphPad Prism version 6 (GraphPad, San Diego, CA, USA). $P$ values $\leq 0.05$ were considered significant.

## Clinical specimen and total RNA isolation

Ten fresh frozen tumour-adjacent normal PTC tissues specimens from UKMMC-UMBI Biobank were subjected to cryosectioning and Haematoxylin and Eosin (H&E) staining. This part of research was approved by the Universiti Kebangsaan Malaysia Research Ethics Committee (UKMREC) (reference: UKM 1.5.3.5/244/UMBI-2015-002). A written

informed consent had been signed by these 10 subjects included in validation phase according to institution's rules and regulations. All the slides were reviewed by the pathologist to assess the percentage of tumour cells and normal cells. Only tumour tissues which contain >80% cancer cells and normal tissues with <20% necrosis were subjected to nucleic acid extraction. Total RNA including miRNA was isolated from the frozen samples using AllPrep DNA/RNA/miRNA Isolation Kit (Qiagen, Hilden, Germany) according to the manufacturer's protocol. The total RNA quality and quantity were assessed via absorbance spectrophotometry on a Nanodrop 1000 instrument (Thermo Scientific, Wilmington, DE, USA) and Qubit$^{TM}$ fluorometer (Invitrogen, USA). Integrity of RNA was assessed using Eukaryote Total RNA Nano chip on Bioanalyzer 2100 (Agilent Technologies, Santa Clara, USA). Only total RNA with RNA Integrity Number (RIN) of at least 6 were used for subsequent steps. Eukaryote Small RNA chip (Agilent Technologies, Santa Clara, USA) was used for determination of concentration and percentage of small RNA.

## Library preparation and next generation sequencing

MiRNA libraries were prepared using Illumina Truseq Small RNA library preparation kit (Illumina, SanDiego, USA) following manufacturer's protocol. Briefly, 3′ and 5′ adapters were sequentially ligated to the ends of small RNAs fractionated from 1 μg of total RNA, and reverse transcribed to generate cDNA. The cDNA was amplified using a common primer complementary to the 3′ adapter, and a primer containing 1 of 48 index sequences. Samples were size-selected (145–160 bp fragments) on a 6% polyacrylamide gel, purified, quantified and pooled for multiplexed sequencing. The resulting pooled libraries were normalized to 2 nM and were hybridized to oligonucleotide-coated single-read flow cells for cluster generation using HiSeq$^®$ Rapid SR Cluster Kit v2 on Hiseq 2500. Subsequently the clustered pooled microRNA libraries were sequenced on the HiSeq 2500 for 50 sequencing cycles using HiSeq$^®$ Rapid SBS Kit v2 (50 Cycle). Base calling was performed using CASAVA (v.1.8.2) (Illumina, San Diego, CA, USA) and short-read sequences in FASTQ format were used for downstream analysis.

## Bioinformatics analyses

The miRNASeq and RNASeq V2 level 3 data from TCGA were used exclusively. The normalised expression (reads per million or RPM) of all miRNAs was log$_2$-transformed and used for fold change calculation. The RNAseq by Expectation-Maximization (RSEM) values (from files with the extension .rsem.genes.results) were used to quantify messenger RNA (mRNA) expression levels. The RSEM algorithm is a statistical model that estimates RNA expression levels from RNA sequencing counts (*Li & Dewey*, *2011*). We then performed the Students' unpaired *t*-test with a Benjamini Hochberg false discovery rate (FDR) multiple testing correction and log$_2$ fold change calculation using Bioconductor version 3.1 (BiocInstaller 1.18.2) (*Gentleman et al.*, *2004*) in R version 3.2.0 (*R Development Core Team*, *2008*) (Files S1 and S2). Downregulated genes will have negative log$_2$ values while upregulated genes will have positive log$_2$ values. Statistical significance is denoted as $p \leq 0.05$. Heatmaps were created using GeneE from the Broad Institute (http://www.broadinstitute.org/cancer/software/GENE-E) while Venn diagrams were

created using Venn online tool (http://bioinformatics.psb.ugent.be/webtools/Venn). All other figures were created or labelled using Adobe Photoshop.

Analysis of the miRNAseq data from our in house experiment were performed using BaseSpace miRNA Analysis app version 1.0.0 (Illumina, San Diego, CA, USA) using the default setting. Briefly, adapters were trimmed using cutadapt, the trimmed reads were mapped on miRNA precursors using SHRiMPS aligner, the reads associated to mature miRNAs were counted and differential expression between experimental conditions were analysed using DESeq2 (*Cordero et al., 2012*). The expression, $\log_2$ fold change and adjusted *p*-value of hsa-miR-146b, hsa-miR-375, hsa-miR-31, hsa-miR-7-2 and hsa-miR-204 were then extracted from the overall results.

## Pathway enrichment analysis and integrated analysis of miRNA and gene expression

The functions and pathways of the differentially expressed genes were annotated and analysed using the annotation tools from the Database for Annotation, Visualization and Integrated Discovery (DAVID) (*Huang, Sherman & Lempicki*, *2009a*; *Huang, Sherman & Lempicki*, *2009b*) according to the steps described in these publications. The identified genes were also jointly annotated against the Kyoto Encyclopedia of Genes and Genomes (KEGG) database (*Kanehisa & Goto*, *2000*). The genes that were annotated in the KEGG database as being involved in signaling pathways were subjected to further analysis. Pathways with Benjamini-adjusted *p* value $\leq 0.05$ were considered to be statistically significant.

Integration of the miRNAs dataset with gene expression dataset and calculation of correlation were performed in MAGIA2, a web tool for the integrated analysis of target predictions, miRNA and gene expression data (*Bisognin et al.*, *2012*). MiRNA target predictions include transcription factor binding sites (TFBS) within miRNA and gene promoters. In this analysis, matched expression data matrices of significantly dysregulated miRNAs and genes (BH adjusted *p* value $\leq 0.05$) were uploaded for integrated analysis. EntrezGene IDs and DIANA-microT (*Maragkakis et al.*, *2009*) target prediction algorithms were selected. Anticorrelated expressions were investigated between miRNA and their putative target genes using Pearson correlation measure.

# RESULTS

## The effect of lymph node status on survival duration of TCGA PTC patients

Overall survival in PTC patients was not influenced by LNM status (Fig. 1A); however, PTC patients with LNM has significantly shorter disease-free survival rate compared to PTC patients without LNM (Log-rank Mantel Cox test, $p = 0.0049$; Fig. 1B).

## Differentially expressed miRNAs

We identified 181 miRNAs which were significantly differentially expressed in PTC LNM-P versus PTC LNN (BH corrected *p* value $\leq 0.05$). Among the 181 miRNAs significantly expressed in PTC LNM-P versus PTC LNN, 110 were upregulated and 71 were downregulated (Table S2). Figure2 illustrates a heatmap representing the expression levels

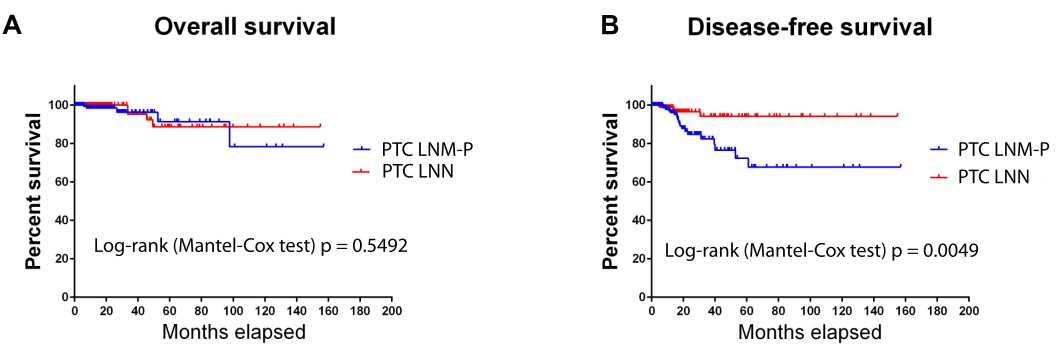

**Figure 1   Survival analysis of PTC with LNM and PTC without LNM.**

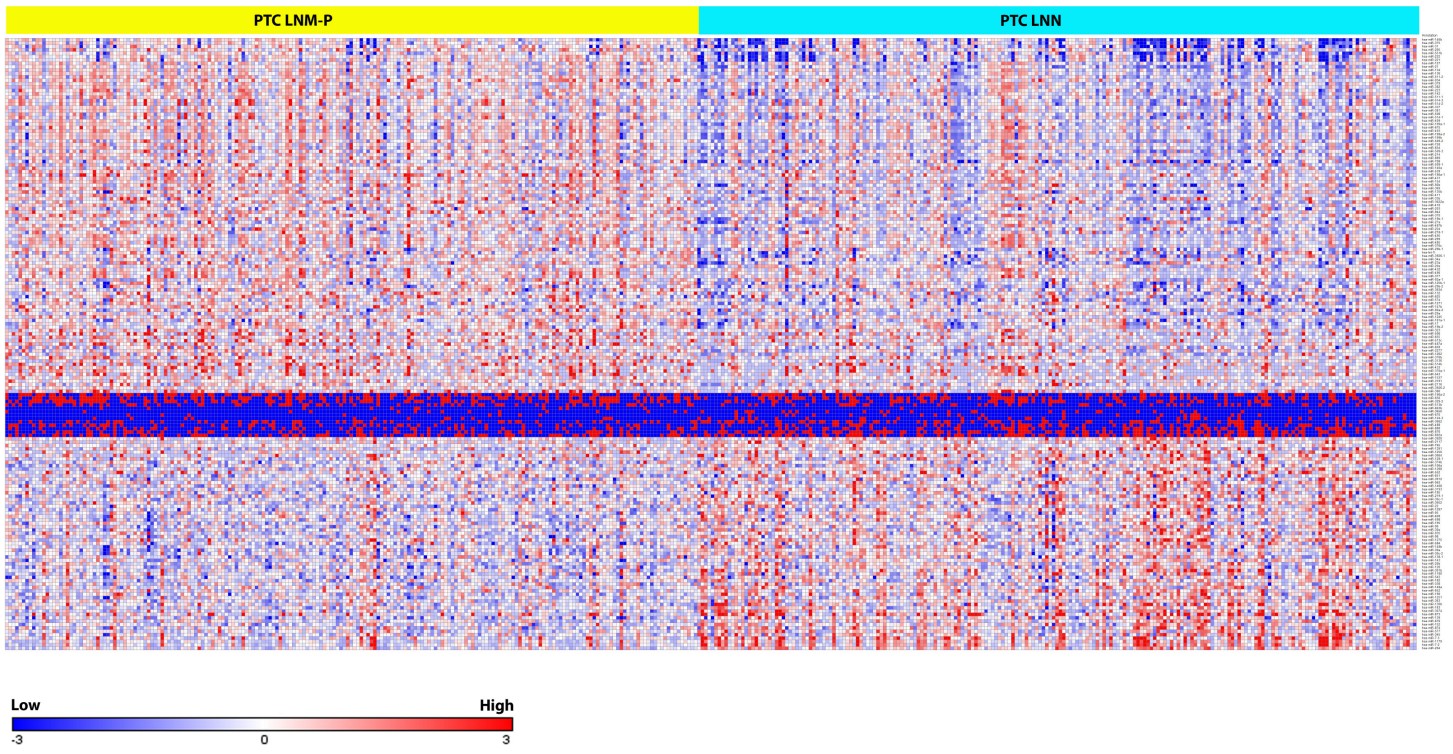

**Figure 2   Heat map of the 181 differentially expressed miRNAs in PTC LNM-P and LNN (Student's *T*-test with BH corrected *p* value ≤ 0.05).**

of 181 deregulated miRNAs in PTC LNM-P versus PTC LNN. The list of top deregulated miRNAs includes hsa-miR-146b, hsa-miR-375, hsa-miR-31, hsa-miR-7-2 and hsa-miR-204 ($\log_2$ fold change 1.7, 1.3, 1, $-1.1$ and $-1.3$, respectively, Fig. 3). On the other hand, 395 miRNAs were differentially expressed between PTC LNM-P and normal thyroid while 400 miRNAs were differentially expressed between PTC LNN and normal thyroid (Figs. S1 and S2, respectively). The list of miRNAs significantly deregulated in PTC LNM-P and PTC LNN compared to normal thyroid is included in Table S3.

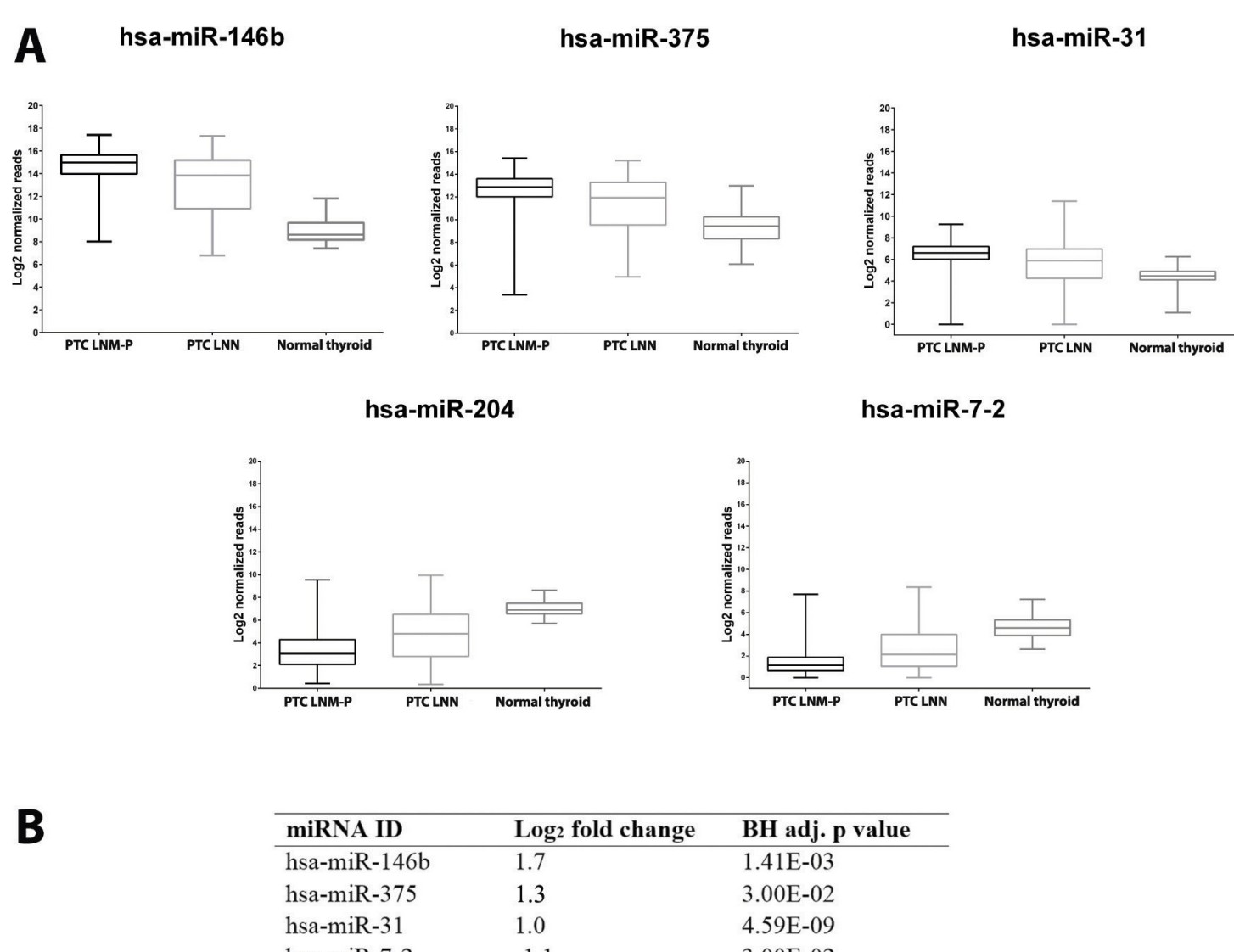

**Figure 3** **Expression levels of five selected miRNAs deregulated in PTC.** Boxplots (A) illustrate log₂ normalized miRNA reads in PTC LNM-P, PTC LNN and normal thyroid. Table (B) showing log2 fold change and *p* value of selected miRNAs in PTC LNM-P compared to PTC LNN.

We then determine the expression of these top deregulated miRNAs in a small set of validation experiment consisted of five pairs of tumour-adjacent normal from each PTC LNM-P and PTC LNN cases (total of 20 samples comprised of five PTC LNM-P, five PTC LNN and 10 adjacent normal thyroid tissues from each patient). As illustrated in Fig. 3B, hsa-miR-146b was significantly upregulated in PTC LNM-P versus adjacent normal thyroid (log₂ fold change 6.0) and in PTC LNN versus adjacent normal thyroid (log₂ fold change 4.7). Similar trends were observed for hsa-miR-375, hsa-miR-31 and hsa-miR-204 in PTC LNM-P versus adjacent normal thyroid (log₂ fold change 3.6, 3.1 and −3.4, respectively, Fig. 3B). However, downregulation of hsa-miR-7-2 in PTC LNM-P

versus adjacent normal thyroid did not reach statistical significance. On the other hand, expression of hsa-miR-375, hsa-miR-7-2 and hsa-miR-204 in PTC LNN versus adjacent normal thyroid were in concordance with our analysis using the TCGA data (log$_2$ fold change 3.6, −2.3 and −2.7, Fig. 3B).

These findings did not deviate much from our analysis using the TCGA PTC data with the exception to hsa-miR-7-2 in PTC LNM-P versus adjacent normal thyroid and hsa-miR-31 in PTC LNN versus adjacent normal thyroid which failed to reach statistical significance (Table S2). This could be explained due to the fact that our validation samples were tumour-adjacent normal tissues while TCGA PTC specimens were of unpaired normal tissues. However, the differential expression of these five top deregulated miRNAs did not reach statistical significant when we compared between PTC LNM-P and PTC LNN (Fig. 3B). This might be due to small sample size in our validation study.

## Differentially expressed genes

Initial filtering revealed 8,611 significantly deregulated genes in PTC LNM-P versus PTC LNN, 14,192 genes in PTC LNM-P versus normal thyroid and 13,392 genes in PTC LNN versus normal thyroid. There were 4,135 upregulated and 4,476 downregulated genes in PTC LNM-P relative to PTC LNN. By increasing the stringency of selection to genes with log$_2$ fold change $\geq 1$ or $\leq -1$, 407 genes were identified as strongly deregulated. Among the strongly deregulated genes were *SFTPB*, *CLDN10*, *DIO1* and *MT1G* (log$_2$ fold change 3.1, 2.9, −2.2 and −2.5 respectively, Table S4). Various cancer-related genes were also differentially expressed significantly, including *BRAF*, *BRCA2*, *VEGFA*, *VEGFB*, *RET*, *PIK3CA*, *CTNNB1* and *GNAS* (Table S4).

## Enriched pathways in PTC LNM-P

The significantly dysregulated genes in PTC LNM-P versus PTC LNN were mainly enriched in 12 KEGG pathways including oxidative phosphorylation (OxPhos), Parkinson's disease, focal adhesion, Alzheimer's disease, valine, leucine and isoleucine degradation, pathways in cancer, cell adhesion molecules (CAMs), leukocyte transendothelial migration, cytokine–cytokine receptor interaction, small cell lung cancer, Huntington's disease and extracellular matrix receptor interaction (Fig. 4A). When we overlapped the results from the three comparison groups (PTC LNM-P versus PTC LNN, PTC LNM-P versus normal thyroid and PTC LNN versus normal thyroid), four unique pathways potentially involved in metastasis to the lymph nodes were significantly enriched, namely, oxidative phosphorylation (OxPhos), cell adhesion molecules (CAMs), leukocyte transendothelial migration and cytokine–cytokine receptor interaction pathways (Fig. 4A). The oxidative phosphorylation pathway was the most significantly perturbed ($p = 4.70E-06$) with general downregulation of 90 OxPhos-related genes (Fig. 5). Focal adhesion and pathways in cancer were commonly enriched in all the three group comparisons. Pathways in cancer is a collection of general cancer-related pathways and is an indication that many essential carcinogenic processes may be under the influence of dysregulated miRNAs (*Pizzini et al.*, *2013*). On the other hand, ECM-receptor interaction pathway and the valine, leucine and isoleucine degradation pathway were commonly enriched only in PTC LNM-P versus PTC

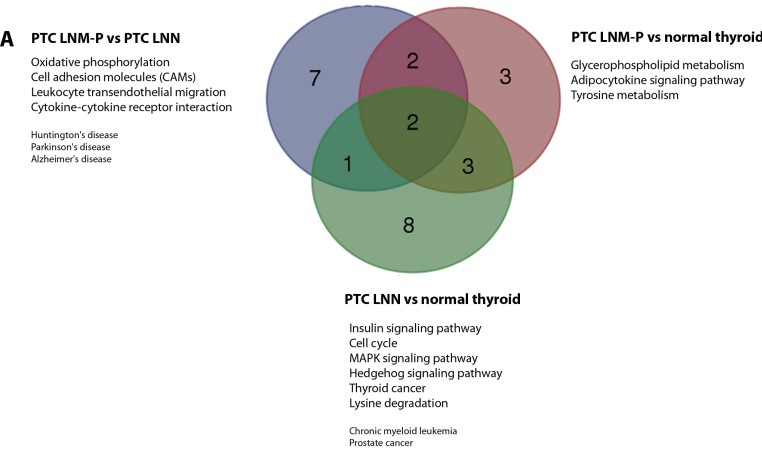

**A** PTC LNM-P vs PTC LNN

Oxidative phosphorylation
Cell adhesion molecules (CAMs)
Leukocyte transendothelial migration
Cytokine-cytokine receptor interaction

Huntington's disease
Parkinson's disease
Alzheimer's disease

PTC LNM-P vs normal thyroid

Glycerophospholipid metabolism
Adipocytokine signaling pathway
Tyrosine metabolism

PTC LNN vs normal thyroid

Insulin signaling pathway
Cell cycle
MAPK signaling pathway
Hedgehog signaling pathway
Thyroid cancer
Lysine degradation

Chronic myeloid leukemia
Prostate cancer

**B**

| Group comparison | Enriched pathway | Benjamini adj. p value |
|---|---|---|
| PTC LNM-P vs PTC LNN | Oxidative phosphorylation | 4.70E-06 |
| | Parkinson's disease | 5.80E-06 |
| | Focal adhesion | 3.40E-05 |
| | Alzheimer's disease | 4.80E-05 |
| | Valine, leucine and isoleucine degradation | 5.50E-03 |
| | Pathways in cancer | 6.00E-03 |
| | Cell adhesion molecules (CAMs) | 7.40E-03 |
| | Leukocyte transendothelial migration | 2.00E-02 |
| | Cytokine-cytokine receptor interaction | 1.90E-02 |
| | Small cell lung cancer | 1.90E-02 |
| | Huntington's disease | 1.90E-02 |
| | ECM-receptor interaction | 3.30E-02 |
| PTC LNM-P vs normal thyroid | ECM-receptor interaction | 8.70E-03 |
| | PPAR signaling pathway | 4.60E-03 |
| | p53 signaling pathway | 4.00E-03 |
| | Pathways in cancer | 5.50E-03 |
| | Focal adhesion | 6.80E-03 |
| | Valine, leucine and isoleucine degradation | 2.10E-02 |
| | Axon guidance | 3.50E-02 |
| | Glycerophospholipid metabolism | 4.60E-02 |
| | Adipocytokine signaling pathway | 5.10E-02 |
| | Tyrosine metabolism | 4.70E-02 |
| PTC LNN vs normal thyroid | Axon guidance | 8.10E-04 |
| | Pathways in cancer | 2.50E-03 |
| | Insulin signaling pathway | 5.30E-03 |
| | Cell cycle | 7.70E-03 |
| | MAPK signaling pathway | 8.10E-03 |
| | Hedgehog signaling pathway | 8.90E-03 |
| | Chronic myeloid leukemia | 8.80E-03 |
| | p53 signaling pathway | 1.80E-02 |
| | Focal adhesion | 2.80E-02 |
| | PPAR signaling pathway | 2.90E-02 |
| | Thyroid cancer | 3.40E-02 |
| | Prostate cancer | 3.30E-02 |
| | Small cell lung cancer | 4.30E-02 |
| | Lysine degradation | 4.10E-02 |

**Figure 4** **Significantly enriched pathways in PTCs.** Significant KEGG pathway associations to 8611 significantly deregulated genes in PTC LNM-P versus PTC LNN, 14,192 genes in PTC LNM-P versus normal thyroid and 13,392 genes in PTC LNN versus normal thyroid.

LNN and PTC LNM-P versus normal thyroid but were not enriched in PTC LNN versus normal thyroid (Fig. 4B).

## Integrated mixed regulatory circuits, involving miRNAs, genes and TFs in PTC LNM-P

To obtain a more comprehensive insight into the molecular circuits behind LNM in PTC, we focused on functional miRNA-target relationships by performing an *in silico* integration between differentially expressed miRNAs and genes using MAGIA2. Transcription factors-miRNA (TF-miRNA) prediction was based on mirGen2.0 database (*Friard et al.*, *2010*) and TransmiR (*Wang et al.*, *2010a*), whereas the TF–gene interactions were acquired from the 'TFBS conserved' track of the University of California Santa Cruz (UCSC) genome annotation for humans (version hg19) (*Bisognin et al.*, *2012*). Our results show that 12 miRNAs are involved in the strongest 200 interactions and they were identified as significant by MAGIA2. Hsa-miR-147b, hsa-miR-301b, hsa-miR-375, hsa-miR-496, hsa-miR-543, hsa-miR-577, hsa-miR-765, hsa-miR-892a, hsa-miR-934, hsa-miR-935, hsa-miR-940 and hsa-miR-944 were predicted to activate or inhibit 3,746 genes and 1,987 TFs (Fig. 6). Hsa-miR-577 and hsa-miR-147b consistently appeared in the top 20 regulatory circuits across all group comparisons. Interestingly, hsa-miR-301b appeared in both of the top 20 circuits in PTC LNM-P versus PTC LNN or normal thyroid but was absent in PTC LNN versus normal thyroid (Fig. S3).

## DISCUSSION

In this study, we explored the landscape of miRNA and mRNA expression of PTC using data obtained from the TCGA THCA project aiming to identify key pathways involved in lymph node metastasis. Our analysis revealed 110 upregulated miRNAs and 71 downregulated

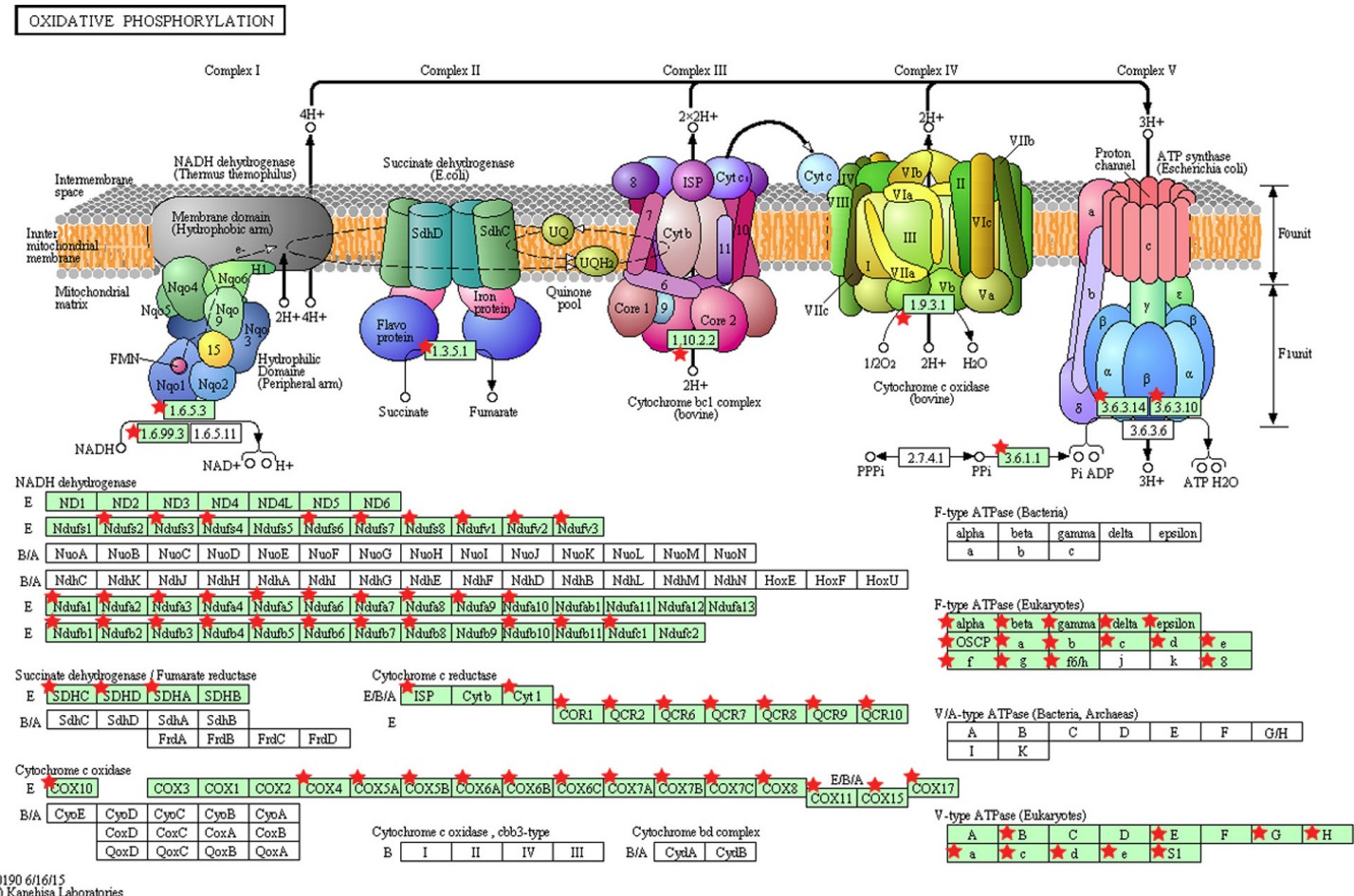

**Figure 5  KEGG pathway map illustrating oxidative phosphorylation in human.** The OxPhos-related genes significantly altered in PTC LNM-P compared to PTC LNN were depicted with red star. Pathway figure was obtained from KEGG (Kyoto Encyclopedia of Genes and Genomes) (*Kanehisa & Goto*, *2000*; *Kanehisa et al.*, *2016*) in DAVID analysis.

miRNAs in PTC LNM-P versus PTC LNN. The top deregulated miRNAs includes hsa-miR-146b, hsa-miR-375, hsa-miR-31, hsa-miR-7-2 and hsa-miR-204. Our findings are supported by several other similar studies, and in particular hsa-miR-146b, which was reported to be upregulated in PTC LNM-P versus PTC LNN (*Lee et al.*, *2013*; *Yang et al.*, *2013*; *Acibucu et al.*, *2014*; *Deng et al.*, *2015*).

Hsa-miR-146 is one of the widely studied miRNAs in thyroid cancers and has been shown to be frequently upregulated in PTC (*He et al.*, *2005*; *Pallante et al.*, *2006*; *Tetzlaff et al.*, *2007*; *Chen et al.*, *2008*; *Yip et al.*, *2011*; *Chou et al.*, *2010*; *Chou et al.*, *2013*; *Sun et al.*, *2013*), anaplastic thyroid cancer (*Fassina et al.*, *2014*) and follicular thyroid cancer (FTC) (*Wojtas et al.*, *2014*). Functional analyses of hsa-miR-146 revealed its involvement in various cellular functions including migration, invasion, proliferation, colony-forming ability, cell cycle, and resistance to chemotherapy-induced apoptosis in *BRAF*-mutated cell lines (*Chou et al.*, *2013*; *Deng et al.*, *2015*; *Geraldo, Yamashita & Kimura*, *2012*). Using multivariate logistic regression analysis, Chou and colleagues, *(2013)* demonstrated that

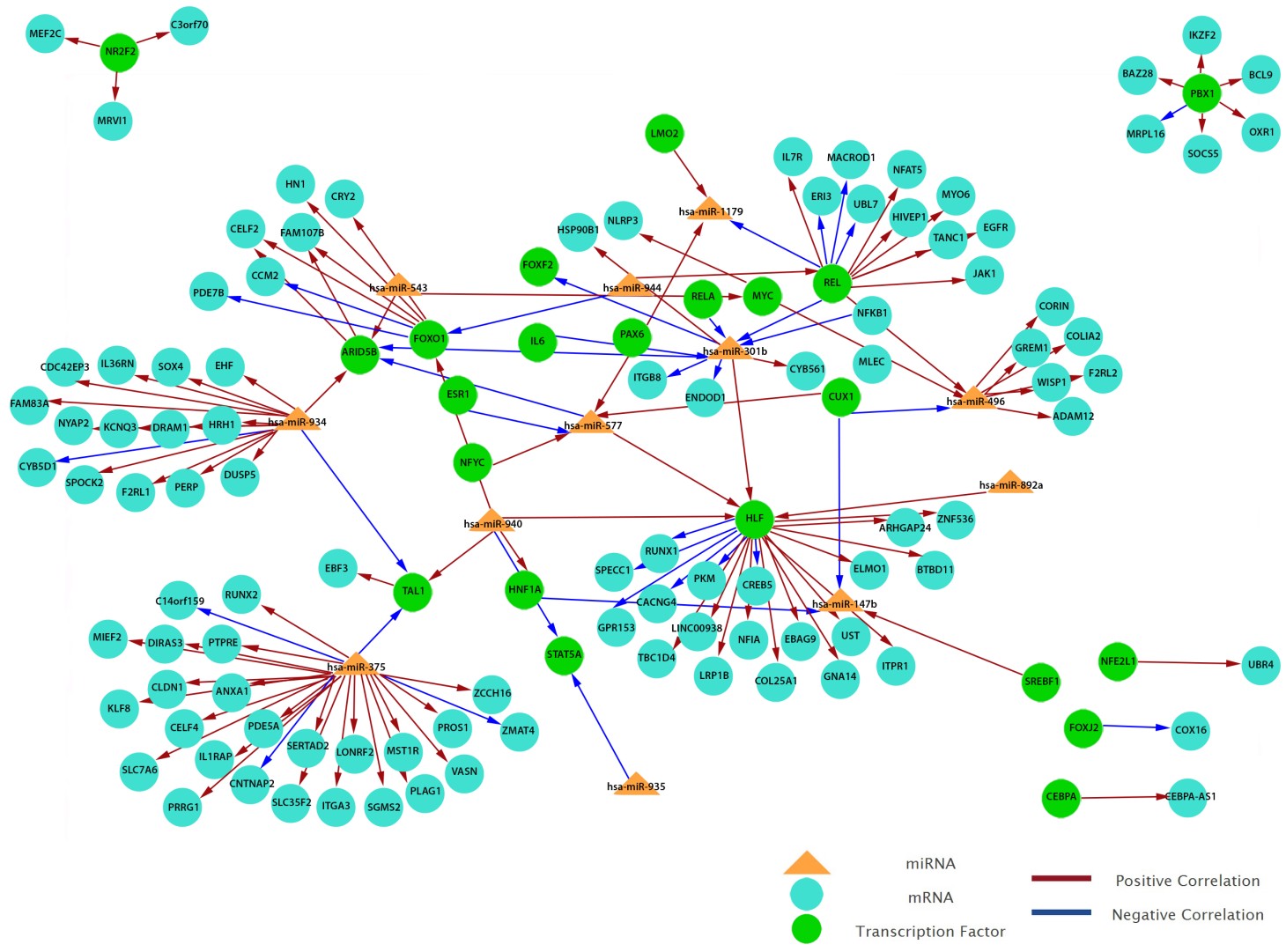

**Figure 6** **Grand view of top 200 regulatory circuits constructed using significantly dysregulated miRNAs and genes in PTC LNM-P compared to PTC LNN.**

increased hsa-miR-146b expression is one of the independent risk factors for poor prognosis in PTC, implicating the potential of this miRNA as a prognostic marker.

The genes targeted by hsa-miR-146b are mostly unknown, and to date there are only two genes which has been reported as the direct targets of this miRNA in PTC. *Geraldo, Yamashita & Kimura* (*2012*) reported *SMAD4*, an important member of the transforming growth factor β (TGF-β) signaling pathway, as the target of hsa-miR-146b-5p. The direct binding of hsa-miR-146b-5p on the *SMAD4* UTR was confirmed via a luciferase reporter assay and the inhibition of hsa-miR-146b-5p expression resulted in significantly increased SMAD4 gene and protein expression levels in the human PTC cell lines. Furthermore, the inhibition of hsa-miR-146b-5p increased the cellular response to the TGF-β anti-proliferative signal, leading to significant reduction of cell proliferation (*Geraldo, Yamashita & Kimura*, *2012*). In a more recent study, the Zinc

Ring Finger 3 (ZNRF3) gene was revealed as a direct target of hsa-miR-146b-5p and this miRNA was shown to stimulate cell migration, invasion and epithelial-to-mesenchymal transition (EMT) by downregulating *ZNRF3* (*Deng et al.*, *2015*). Another study showed that *ZNRF3* inhibits Wnt signaling by interacting with FZD and LRP 5/6 complexes, hence promoting Wnt receptor ubiquitination and degradation (*Hao et al.*, *2012*). Hsa-miR-146b-5p increases the cell surface levels of FZD6 and LRP6 via suppression of *ZNRF3*, causing enhanced Wnt/$\beta$-catenin signaling. These findings revealed a novel mechanism of hsa-miR-146b-5p in mediating the induction of EMT and implied the role of *ZNRF3* as a tumor suppressor in PTC (*Deng et al.*, *2015*). Additional efforts to identify genes controlled by hsa-miR-146b associated with LNM will eventually revealed new biomarkers that can be utilized to correlate with disease outcome in PTC patients.

Hsa-miR-204 expression in PTC LNM-P is significantly lower than in PTC LNN. This is the first report showing the downregulation of hsa-miR-204 in PTC LNM-P. This miRNA was also downregulated in PTC compared to adjacent normal thyroid tissue and noncancerous thyroid (*Swierniak et al.*, *2013*). It is likely that hsa-miR-204 is downregulated in PTC compared to normal or benign thyroid disease and is further supressed when lymph node metastasis occurs. This miRNA is known as a tumor suppressor miRNA and is downregulated in various cancers including renal clear cell carcinoma (*Gowrishankar et al.*, *2014*), minimal deviation adenocarcinoma (MDA) of uterine cervix (*Lee et al.*, *2014*) and breast cancer (*Li et al.*, *2014*). This miRNA has also been shown to have a prognostic value; low level of hsa-miR-204-5p expression was correlated with LNM, advanced stage and low survival rate in endometrial cancer (*Bao et al.*, *2013*), and also poor prognosis in colorectal cancer (*Yin et al.*, *2014*). *In vitro* functional analyses revealed the involvement of hsa-miR-204 in inhibiting the clonogenic growth, migration and invasion of endometrial carcinoma cells (*Bao et al.*, *2013*). In addition, restoration of hsa-miR-204-5p expression supressed cell proliferation, migration, invasion and induced apoptosis and chemotherapeutic sensitivity in colorectal cancer cell (*Yin et al.*, *2014*).

The validated targets for hsa-miR-204 in PTC are also not well-characterized. To date there is only one study investigating the functional role of hsa-miR-204 in PTC (*Liu et al.*, *2015*). Enforced expression of hsa-miR-204-5p inhibited cell proliferation and induced apoptosis and cell cycle arrest in PTC cell lines (TCP-1 and BCPAP). In addition, hsa-miR-204-5p also inhibits PTC cell tumorigenicity *in vivo* (*Liu et al.*, *2015*). Bioinformatics prediction analyses using three algorithms (miRanda, Pictar, and TargetScan) revealed the insulin-like growth factor-binding protein 5 (IGFBP5), a gene playing an essential role in carcinogenesis (*Beattie et al.*, *2006*), as a potential target of hsa-miR-204-5p. Luciferase reporter assay confirmed the direct binding of hsa-miR-204-5p to the 3′ UTR of *IGFBP5* (*Liu et al.*, *2015*). In the same study, hsa-miR-204-5p and *IGFBP5* expression were also shown to be inversely correlated. Their findings confirmed the role of hsa-miR-204-5p as a tumor suppressor in PTC and revealed the potential use of this miRNA as a therapeutic agent in the treatment of PTC.

In an attempt to identify genes and pathways associated with mortality in PTC, *Nilubol et al.* (*2011*) performed genome-wide expression (GWE) analysis in 64 PTC patients and identified the oxidative phosphorylation pathway as one of the significantly perturbed

pathways. In addition, Lee and colleagues, *(2015a)* also showed that the expression of OxPhos gene sets was significantly lower in primary PTC than in matched normal thyroid tissue. Our findings revealed a similar trend with OxPhos genes being significantly downregulated in PTCs versus normal thyroid tissues as well as in PTC LNM-P versus PTC LNN. However, significant enrichment of OxPhos pathway was only observed in PTC LNM-P compared to PTC LNN. Alteration in metabolic processes has been considered as an indispensable component of malignant transformation (*Lee et al.*, *2015a*) thus the involvement of oxidative phosphorylation in LNM in PTC necessitates further investigation.

Oxidative phosphorylation is a process whereby an adenosine triphosphate (ATP) is produced as a result of electrons transfer from nicotinamide adenine dinucleotide (NADH) or flavin-adenine dinucleotide ($FADH_2$) to oxygen by a series of electron carriers (*Berg, Tymoczko & Stryer*, *2002*). The thyroid gland is an endocrine organ with a high energy consumption and oxidative processes are crucial for thyroid hormone synthesis (*Lee et al.*, *2015b*). The mitochondria is responsible for providing 90% of the cellular energy necessary for various biological functions through oxidative phosphorylation and plays an important role in energy metabolism in the normal thyroid gland and in thyroid tumors (*Kim et al.*, *2012*). The mitochondria is involved in many cell signaling pathways by playing crucial roles in apoptosis, cell proliferation and cellular $Ca^{2+}$ homeostasis (*Rustin*, *2002*). Mitochondrial DNA (mtDNA) content was shown to be higher in PTC compared to the paired normal DNA and in normal controls (*Mambo et al.*, *2005*). Despite advancement in the elucidation of molecular events underlying thyroid carcinogenesis in the last decade, the function and nature of energy metabolism in thyroid cancer remain unclear (*Lee et al.*, *2015b*).

In addition to oxidative phosphorylation, we also identified significant enrichment of other cancer-related pathways such as cell adhesion molecules (CAMs), leukocyte transendothelial migration and cytokine–cytokine receptor interaction pathways which were unique to PTC LNM-P versus PTC LNN. Interestingly, these pathways were not significantly enriched when PTCs (LNM-P and LNN) were compared to normal thyroid tissues. Taken together, it could be hypothesized that metastasis to the lymph node in PTC occurred via changes in the aforementioned pathways. However, some pathways in our analysis, such as valine, leucine and isoleucine degradation, could not be associated with oncogenesis or metastasis and may need further investigation.

Our integrated analysis revealed hsa-miR-301b's presence in the top 20 circuits in both PTC LNM-P versus PTC LNN and PTC LNM-P versus normal thyroid but was absent in PTC LNN versus normal thyroid despite significant downregulation with modest fold change ($\log_2$ fold change of −0.3). Hsa-miR-301 is located in the intronic region of *SKA2* (spindle and kinetochore associated complex subunit 2) and belongs to the hsa-miR-130 microRNA precursor family (*Cao et al.*, *2010*). In contrast to our findings, hsa-miR-301 upregulation has been reported in various cancers of non-thyroid origins and given that a miRNA can act either as an oncomiR or tumor suppressor depending on the cellular context and tissue type (*Garzon, Calin & Croce*, *2009*), this observation is not unexpected. There is no evidence of hsa-miR-301b dysregulation in PTC so far, but it was reported to be upregulated in follicular thyroid adenoma compared to normal thyroid tissue (*Rossing et al.*, *2012*). It is also upregulated in CRC without LNM in comparison to paracancerous

control (*Wang et al.*, *2010b*). The inhibition of hsa-miR-301 decreased breast cancer cell proliferation, clonogenicity, migration, invasion, tamoxifen resistance, tumor growth and microvessel density, further establishing this miRNA as an oncomiR (*Shi et al.*, *2011*). *FOXF2*, *BBC3*, *PTEN*, and *COL2A1* were confirmed as its direct targets through luciferase reporter assays (*Shi et al.*, *2011*).

Transcription factors (TFs) are a group of proteins involved in the initiation of transcription and are important for the regulation of genes. Majority of oncogenes and tumor suppressor genes encode the TFs (*Ell & Kang*, *2013*). Dysregulation of oncogenic or tumor suppressive TFs could influence multiple steps of the metastasis cascade, leading to cancer progression (*Ell & Kang*, *2013*). The involvement of TFs in PTC has been investigated since decades ago and several thyroid-specific TFs have been identified (*Guazzi et al.*, *1990*; *Fabbro et al.*, *1994*). Most recently, the glioma-associated oncogene homolog 1 (GLI1) has been identified as a TF marker for LNM in PTC and it increases tumor aggressiveness via the Hedgehog signaling pathway (*Lee et al.*, *2015c*). The hepatic leukemia factor (HLF) is the only TF which appeared in the top 20 circuits of PTCs with or without LNM versus normal thyroid from our integrated analysis. On the other hand, REL was identified in the top 20 circuits only in PTC LNM-P in comparison to PTC LNN and will be discussed further in the following section.

The HLF is a transcription factor that facilitates thyroid hormone activation from the thyroid hormone receptor/retinoid X receptor heterodimer to hypoxia-inducible factor (HIF-1$\alpha$) (*Otto & Fandrey*, *2008*). Triiodothyronine (T3) indirectly increases HIF-1$\alpha$ mRNA by increasing the expression of HLF, subsequently initiating the transcription of HIF-1$\alpha$ transcription factor (*Burrows et al.*, *2011*). HIF is another transcription factor which acts under hypoxia and thus is active in a number of diseases associated with low oxygen environment including cancer (*Burrows et al.*, *2011*). In fact, the HIF-1$\alpha$ protein was differentially expressed in primary thyroid cancers associated with advanced stage; its expression was supressed in normal thyroid tissue and was highest in the most aggressive dedifferentiated anaplastic thyroid carcinomas (ATCs) (*Hanada, Feng & Hemmings*, *2004*), supporting its role for thyroid tumor aggressiveness, progression as well as metastasis. In addition, we also identified a significant involvement of REL/NFkB in lymph node metastasis of PTC which is in concordance with previously published data (*Du et al.*, *2006*).

In summary, we found evidence of five miRNAs differentially expressed in PTC LNM-P. Enrichment analysis revealed that alteration in oxidative phosphorylation pathway could be a key event involved in the lymph node metastasis of PTC suggesting that manipulation of the energy metabolism processes may provide an alternative therapeutic target for tackling metastasis or recurrence. In addition, via the integrated analysis we discovered that hsa-miR-301b might be involved in promoting LNM in PTC via activation of HLF, HIF and REL/NFkB. As far as we know, the roles of these TFs have been explored in PTC; however the exact roles of this miRNA with these TFs in LNM in PTC have not been studied. Hence, further investigation is necessary for future research in order to completely unravel the mechanism of LNM in PTC.

### Funding

This manuscript was supported by the Fundamental Research Grant Scheme (FRGS) from the Ministry of Education Malaysia (FRGS/1/2014/SKK01/UKM/03/1). The funders had no role in study design, data collection and analysis, decision to publish, or preparation of the manuscript.

### Grant Disclosures

The following grant information was disclosed by the authors:
Fundamental Research Grant Scheme (FRGS): FRGS/1/2014/SKK01/UKM/03/1.

### Competing Interests

The authors declare that there are no competing interests.

### Author Contributions

- Nurul-Syakima Ab Mutalib conceived and designed the experiments, performed the experiments, wrote the paper, prepared figures and/or tables.
- Sri Noraima Othman analyzed the data, contributed reagents/materials/analysis tools.
- Azliana Mohamad Yusof analyzed the data.
- Shahrun Niza Abdullah Suhaimi, Rohaizak Muhammad and Rahman Jamal wrote the paper, reviewed drafts of the paper.

### Human Ethics

The following information was supplied relating to ethical approvals (i.e., approving body and any reference numbers):
Universiti Kebangsaan Malaysia Research Ethics Committee (UKMREC).

### Ethics

The following information was supplied relating to ethical approvals (i.e., approving body and any reference numbers):
Universiti Kebangsaan Malaysia Research Ethics Committee (UKMREC) (reference: UKM 1.5.3.5/244/UMBI-2015-002).

### Data Availability

The raw data generated from the validation study has been supplied as Data S1.

### Supplemental Information

Supplemental information for this article can be found online at http://dx.doi.org/10.7717/peerj.2119#supplemental-information.

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
