# Peer review of "Integrated microRNA, gene expression and transcription factors signature in papillary thyroid cancer with lymph node metastasis"

_PeerJ, doi:10.7717/peerj.2119_

## Round 0.1 · original submission · Major Revisions

Please address the comments from three reviewers especially about the validity of your results, and the robustness of the study.

Reviewer 1 ·

Basic reporting

The submission adhere to all PeerJ policies.

Experimental design

No wet experiments were done. The submission was based purely on public data from TCGA.

The data analysis protocol seems solid with main stream bioinformatics tools, such as DAVID, KEGG, etc.

The analysis of different data types (miRNA, gene expression) were conducted independently, rather than an integrated way.

Validity of the findings

Although the findings are obtained based public data and popular bioinformatics tools, i.e., these findings are reproducible in a technical way; however, there were limited validation from other approaches. Subjectively, I would not believe the findings without additional evidence.

The article is basically a TCGA data analysis project. It is very difficult for me to judge the value of these findings from clinical perspective, maybe it is better to redirect this article to a more specialized journal related to papillary thyroid cancer.

Additional comments

N/A

Reviewer 2 ·

Basic reporting

The manuscript used TCGA PTC data, including miRNA, mRNA together with transcription factors, through integrated analysis, it declared improved identification of functional miRNA-target gene-TF relationships. And it finally concluded that "We found evidence of five miRNAs differentially expressed in PTC LNM-P. Alteration in OxPhos pathway could be the central event in metastasis to the lymph node in PTC"

Experimental design

The methods used in the manuscript is just routine and not novel at all. Base on t-test and folder change, it first choose the top miRNA and genes, then submit to MAGIA, a online tool, for integrated analysis, then pathways were generated.
1. Figure 1, what is overall survival and disease-free survival, there is no description.
2. Figure 2, The heat map was very confused, there was a "dark blue belt" with lots of dark red dots, I guess these are less than 3 and larger than 3, which would be outliers. The authors should check the data quality. Further, this "dark blue belt" would violate the Gaussian Distribution in T-test assumption, so the p-value is not precise.
3. Figure 3. has-miR-31 has the smallest p-value, but its box plot shows the least difference. please double check.

Validity of the findings

The manuscript found top miRNA and pathways just based on TCGA PTC data only, without any replication, so the conclusion is not convincing besides the data quality and methods issues..

Reviewer 3 ·

Basic reporting

The paper presents a study about miRNA and mRNA expression of PTC using data from the TCGA project. The authors' aim to identify key pathways involved in lymph node metastasis.
Authors present a good analysis of the problem as well as interesting discussions on pathway enrichment in different subtypes of Papillary thyroid carcinoma.
While the work is presented in with a good storyline, the supplementary material does not include all the details necessary to reproduce the study results.
It is very important that the authors address this critical issue related to the the reproducibility crisis in the scientific field.

More details of specific parts of the paper are listed below.

Experimental design

* Authors says "data was obtained from the TCGA" and they provide the link to the TCGA website. However no specific information about the query parameter is given. Please provide detailed information about all the parameters to allow future researcher to download exactly the same type of data for reproducibility purposes.

* While plots and output tables of analysis are useful for the purpose of the paper, it is crucial to provide the code to reproduce the results. Sentences like "statistical analyses were perfomed using..., we performed the test using bioconductor..., Heatmaps were created..., genes were annotated and analysed using..." should be accompanied by source code to reproduce the analysis or extremely detailed description of parameters/steps used to obtain those results.

Minor issues:

* in Figure 3 there are two values are 1.4 in the table, but they are listed as 1.3 in the text and in the supplementary table.

* in Figure 6. it is not clear to me what the grey element in the legend (selected relation) is. It doesn't seem to appear in the drawing as all connections seem to be either positive or negative correlations. Adding more spaces between some crowded clouds could also improve readability.

Validity of the findings

While the discussion seems reasonable, the missing details (stated above) do not allow me to comment on the specific robustness, or soundness of the study.

---

## Round 0.2 · Minor Revisions

I believe that the English language still needs work.

Please, delete irrelevant "general" phrases and sentences. Delete unneeded words. Simplify. Use short sentences. Some Introductory sentences are irrelevant or are not needed. I suggest you shorten the MS, especially the Discussion.

Try to avoid vague words: regulate, modulate, affect, manipulate, interact, play role, target, closely related, dependent, mediated, impact.
Instead use concrete words: increase, decrease, stimulate/activate, inhibit/suppress, bind, correlate. For example, instead of "regulate" use either increase or decrease, activate or inhibit.

Reviewer 3 ·

Basic reporting

The authors provided answers to all my previous concerns, shared source code for R scripts and provided description of their process that allows people in the future to reproduce their results.

The only minor issue is that one of the figure (Figure 6) is pretty hard to read and the authors say the they "were unable to add more spaces between the clouds because the figure appeared by default from the analysis". While I know that these figures are generated by the online tool, every time I tried to generate those figures from MAGIA2, I obtained pretty well spaced graphs. The cumbersomeness of the authors' figure may be due to a browser resolution or zooming issue, or maybe it was generated a while ago and maybe recent changes in the MAGIA platform fixed bugs that created the overlapping figures. If they didn't do it already, I encourage the author to try to regenerate the figure: it's not a major issue, but proper data visualization is a key feature in science as it may help understanding underling biological mechanisms.

Experimental design

No Comments

Validity of the findings

No Comments

---

## Round 0.3 · accepted · Accept

Please double check reference ordering and reference formats.